# Defining Energy-Dense, Nutrient-Poor Food and Drinks and Estimating the Amount of Discretionary Energy

**DOI:** 10.3390/nu14071477

**Published:** 2022-04-01

**Authors:** Anja Biltoft-Jensen, Jeppe Matthiessen, Karin Hess Ygil, Tue Christensen

**Affiliations:** National Food Institute, Technical University of Denmark, 2800 Kongens Lyngby, Denmark; jmat@food.dtu.dk (J.M.); karinhy@outlook.dk (K.H.Y.); tuchr@food.dtu.dk (T.C.)

**Keywords:** nutrient profiling, dietary guidelines, dietary pattern, recommended diet

## Abstract

Overconsumption of energy provided by energy-dense, nutrient-poor (EDNP) food and drinks increases the risk of unhealthy weight gain and of obesity-related health outcomes. The aim of this study was to develop a nutrient profiling model for classifying EDNP food and drinks and to estimate the amount of discretionary energy for EDNP food and drinks in a recommended diet. A stepwise approach was used first to develop a nutrient profiling model for classifying EDNP food and drinks and then to estimate the amount of discretionary energy in a recommended diet using diet modeling. The nutrition profiling model comprised 24 macro- and micronutrients and energy density. The model classified 67% of 1482 foods and 73% of 161 drinks correctly as EDNP food and drinks compared with an expert-adjusted model. Sweets, chocolate, cake, cookies and biscuits, sweet and salty snacks, sugar-sweetened and artificially sweetened drinks, and alcoholic drinks were classified as EDNP food and drinks. The estimated amount of discretionary energy for EDNP food and drinks was 4–6% of the energy requirements for 4–75-year-old Danes. It seems prudent to have special attention on EDNP food and drinks in dietary guidelines and future public health initiatives to avoid overconsumption of energy.

## 1. Introduction

Findings from the Global Burden of Disease Study 2015 show that the prevalence of overweight and obesity among children and adults is a large and increasing public health problem around the world [1]. One aspect of dietary risk is the overconsumption of energy-dense, nutrient-poor (EDNP) food and drinks. Guidance on EDNP food and drinks, especially in Western countries, is of importance because of the high consumption and the high prevalence of overweight and obesity, and obesity-related diseases such as type 2 diabetes and some cancers [2,3,4,5,6,7].

Data from Euromonitor 2018 show that people in Western countries have a high consumption of EDNP food and drinks, and Denmark holds the world record in consumption of candy. On average, people in the Scandinavian countries, the Netherlands, the USA, England, and Italy buy between 32 and 51 kg candy, chocolates, cake, ice cream, dessert, and snack bars each year [8]. In these countries, people buy between 72 and 186 L of sweet drinks annually. If the amount of sweet foods (35 kg) and sweet drinks (127 L) Danes buy annually is converted to energy, it is equivalent to every Dane obtaining approximately 20% of their daily energy intake from EDNP food and drinks. In the Euromonitor data, homemade cakes and desserts, duty-free and illegal trade of candy and chocolates, and what is bought in canteens, cafés, and restaurants are not included [8]. Therefore, the real consumption may be higher.

Despite the relevance of reducing the intake of EDNP food and drinks, there exists no internationally agreed definition of EDNP food and drinks or how to classify these food and drinks. They are often referred to as food and drinks that are high in fat, added sugars, sodium, and alcohol while containing low levels of nutrients. In the Dietary Guidelines for Americans 2020–2025, they are referred to as “Limit Foods and Drinks Higher in Added Sugars, Saturated Fat, and Sodium, and Limit Alcoholic Drinks” [9], and in Australia as discretionary foods not included in the five major food groups (dairy, fruit, grains, meat and beans, and vegetables) and as food and drinks with higher fat/sugar/alcohol and a lower overall nutrient density [10]. The USA, Australia, and Denmark have estimated the maximum energy intake from EDNP food and drinks that would be possible to consume when complying with nutrition recommendations and dietary guidelines to ensure optimal nutrition [3,9,10,11]. In the USA, the Dietary Guidelines Advisory Committee has estimated an amount of energy that can be used for EDNP food and drinks called discretionary energy that is 6–18 E% of the total energy requirement [3]. In Australia, discretionary energy is approximately 4–14% of the total energy requirement [10]. Denmark has previously estimated discretionary energy in a recommended diet for children and adolescents at approximately 5–7% of the total energy requirements [12]. Since then, new dietary guidelines and nutrient recommendations have been published, and the diet has changed in Denmark. These factors influence the amount of discretionary energy available in a recommended diet.

For people themselves to be able to limit the intake of discretionary food and drinks, it is essential that they know which food and drinks are classified as discretionary, and the maximum amount of EDNP food and drinks that may be included in a recommended diet.

The aim of this study was therefore, first, to develop a nutrient profiling model for classifying EDNP food and drinks and then to estimate the amount of discretionary energy for EDNP food and drinks in a recommended diet to inform official dietary guidelines and future public health initiatives.

## 2. Materials and Methods

First, a nutrient profiling model, including 24 macro- and micronutrients and energy density, was developed and used to score 1482 foods and 161 drinks in an attempt to classify EDNP food and drinks. Second, the classification of EDNP food and drinks was used in modeling recommended diets and estimating the amount of discretionary energy. The study design is illustrated in Figure 1.

Four key terms were used in the current paper: EDNP food and drinks were defined as food and drinks that contain a lot of energy per serving but few nutrients. Core food and drinks were defined as food and drinks that, in appropriate amounts, contribute to a healthy and varied diet that satisfies the nutritional needs and supports overall good health. Discretionary energy was defined as the maximum amount of energy that can be used for EDNP food and drinks in a recommended diet. A recommended diet was defined as a healthy and varied diet that meets the Nordic Nutrition Recommendations 2012 and the official Danish Dietary Guidelines 2013 [7,14].

### 2.1. Development of a Nutrient Profiling Model to Classify Energy-Dense, Nutrient-Poor Food and Drinks

The method for developing the nutrient profiling model consisted of three overall steps summarizing the approach suggested by Scarborough et al. (2007) [15] and the WHO (2010) [16]: (1) conceptual model development and score allocation; (2) setting cut-off and classification of food and drinks; (3) model validation.

The development of the model involved testing 1482 foods and 161 drinks from a comprehensive food list. The food list was developed for the most recent Danish National Survey of Diet and Physical Activity (DANSDA 2011–2013) and therefore contains a wide selection of the most common food and drinks that Danes consume. The list of food and drinks includes both single foods and composite foods such as burgers and smoothies. Foods and drinks were scored separately, as drinks generally have a much lower nutrient and energy density than foods. Information about nutrient content originates from analyzed data retrieved from the Frida database 4, which was published on 2 August 2019 [17].

#### 2.1.1. Step 1: Conceptual Model Development and Score Allocation

We used the across-the-board approach to score food and drinks in absolute terms as the aim was to assess the nutrient profile of each food or drink. As no single food contains all nutrients in the quantities that a person needs, the authors decided to include a wide range of qualifying and disqualifying nutrients in the model. These include 24 macro- and micronutrients where reliable food data have been published (Figure 1). Furthermore, energy density was included in the model. The nutrient density score was determined by the ratio of nutrient content in a food/drink relative to the daily reference values for “the most demanding person”, and this includes younger children with a low energy requirement of around 6 MJ per day as described in the Nordic Nutrient Recommendations 2012 [7]. Individual scores were calculated for each nutrient in a food or drink. The minimum score was 0 and the maximum was 100 for each nutrient. Scores for qualifying and disqualifying nutrients were calculated to estimate the mean score of qualifying and disqualifying nutrients. The higher the score, the more favorable the nutrient density in a food or drink.

Energy density is defined as energy per gram of a food or drink and may be used as an indicator of the risk of overconsumption of energy. The cut-off value for energy density for foods was set at 900 kJ/100 g, corresponding to a high energy density according to the World Cancer Research Fund/American Institute for Cancer Research [18]. Foods with an energy content of 900 kJ/100 g and above scored linearly less up to an energy density of 1800 kJ/100 g, where the score was 0. Results below 0 and above 100 were set to 0 and 100, respectively. The cut-off value for drinks was set at 90 kJ/100 g. This value has been used in the Nutri-Score model, as drinks with an energy density over 90 kJ/100 g indicate a lower nutritional quality [19]. Drinks with an energy content of 90 kJ/100 g and above scored linearly less up to an energy density of 180 kJ/100 g, where the score was 0.

We decided to weigh qualifying and disqualifying nutrients and energy density equally in the model, i.e., with ⅓ each, since it was not possible to find scientific evidence for prioritizing the scores for qualifying and disqualifying nutrients and energy density. The model calculation is illustrated in Appendix A.

The actual calculations of the scores were performed in Microsoft Excel 2016 and checked in SAS, SAS Institute Inc. Cary, NC, USA (Version 9.04).

#### 2.1.2. Step 2: Setting a Cut-Off Value and Classification of Food and Drinks

The cut-off value for classifying EDNP food and drinks was selected pragmatically by the nutrition expert group. The expert group consisted of the authors who have expertise with the development of nutritional profiling models and estimation of discretionary energy using dietary modeling [12,20]. To determine the cut-off value, a literature search was carried out to explore which cut-offs have been used in other nutrient profiling models and how they were determined. The search found no papers to determine the cut-off used for the present model, and therefore no additional information was available to substantiate the decision.

#### 2.1.3. Step 3: Model Validation and Statistics

After the classification of all food and drinks with the nutrient profiling model, quantitative criteria of agreement between the expert-adjusted model and the nutrient profiling model were estimated. After classifying food and drinks with the nutrient profiling model, exceptions and reclassifications of food and drinks were made. This resulted in an expert-adjusted model for food and drinks.

Model validity was measured by sensitivity and specificity analyses. These analyses were performed with the selected cut-off value. Core food and drinks with a score above the cut-off value were considered true positives, while those with a score less than or equal to the cut-off value were considered false negatives. Similarly, EDNP food and drinks below the cut-off value were true negatives, while those above were false positives. The number of food and drinks in these four categories is the predicted classification.

The classification of food and drinks by the expert-adjusted model was included as a response variable, and the nutrient profiling model score was included as a prediction variable.

The sensitivity ratio aims to test if the proportion of food and drinks classified as core foods or drinks by the profiling model also belong to the food and drinks classified as core by the expert-adjusted model. The specificity ratio aims to check if the food and drinks classified as EDNP by the profiling model also belong to the food and drinks classified as EDNP food and drinks by the expert-adjusted model. An ideal nutrient profiling scheme will result in a sensitivity ratio and a specificity ratio equal to one.
$$\begin{array}{c} \mathrm{Sensitivity}=\frac{\mathrm{Nuber}\;\mathrm{of}\;\mathrm{true}\;\mathrm{positives}}{\left(\mathrm{Number}\;\mathrm{of}\;\mathrm{true}\;\mathrm{positives}\;+\;\mathrm{Number}\;\mathrm{of}\;\mathrm{false}\;\mathrm{negatives}\right)}\\ \mathrm{Specificity}=\frac{\mathrm{Nuber}\;\mathrm{of}\;\mathrm{true}\;\mathrm{negatives}}{\left(\mathrm{Number}\;\mathrm{of}\;\mathrm{true}\;\mathrm{negatives}\;+\;\mathrm{Number}\;\mathrm{of}\;\mathrm{false}\;\mathrm{positives}\right)} \end{array}$$

Finally, the receiver operating characteristic (ROC) curve was used to investigate the cut-off value and balance between false negatives and false positives. Key parameters were calculated in IBM SPSS Statistics, IBM Corp., New York, NY, USA version 25.

### 2.2. Estimating the Amount of Discretionary Energy for EDNP Food and Drinks

To examine how much energy is available for EDNP food and drinks in a recommended diet, dietary intake patterns were modeled to provide suggestions for healthy and nutritionally appropriate diets at different energy levels. The modeling was conducted on the eaten diet and not the raw ingredients as the Nordic Nutrition Recommendations 2012 refer to nutrient intake [7]. The diet modeling was conducted in five steps inspired by Davis et al. [11].

#### 2.2.1. Step 1: Setting Overall Criteria and Food and Nutrient Goals for the Recommended Diets

Criteria were that the recommended diets should meet the Nordic Nutrition Recommendations 2012 [7] and the official Danish Dietary Guidelines 2013 [14] and that the modeling data should be intake data from DANSDA 2011–2013. The survey contains intake data from 3946 4–75-year-olds randomly drawn from the Danish Civil Registration System comprising non-institutionalized, free-living Danish citizens [21]. The official Danish Dietary Guidelines from 2013 were used as a dietary goal [6].

Furthermore, the goal of the recommended diets was the recommended intake of dietary fiber and macronutrients as a percentage of the total energy intake (E%) and an intake of 10 vitamins and 8 minerals corresponding to the Recommended Intake (RI) or higher. The recommended diets were designed for use by individuals in specific age and sex groups. The RI is intended to be used for diet planning for groups of individuals in specific age and sex groups [7].

#### 2.2.2. Step 2: Defining Age and Sex Groups

Age and sex groups for the recommended diets were determined from the variation in energy requirements according to the Nordic Nutrition Recommendations 2012 and taking into account:The number of age groups necessary when considering differences in the energy and nutrient requirements between groups.The variation in energy requirements within an age and sex group constitutes a maximum of 10% (maximum 1 MJ/day) compared to the average energy requirement in the age and sex groups.A sufficient number of participants from DANSDA 2011–2013 for the diet modeling.

The energy requirement in each age and sex group was set as the average of the lowest and highest energy requirements within the group and was based on a moderate activity level determined by pedometry in the Danish population [22].

#### 2.2.3. Step 3: Developing Main Food Groups and Subgroups

Each food or drink group was based on actual intakes in Denmark and composed of the same proportions as they are eaten or drunk by different sexes and age groups. The nutrient profiles for each composite food group were therefore different depending on age and sex. A total of 38 food and drink groups were formed.

#### 2.2.4. Step 4: Developing the Recommended Diets

The recommended diets were constructed by performing a nutrition calculation for the different age and sex groups based on the composite food and drink groups and the amounts of food and drinks stated in the dietary guidelines per 10 MJ adjusted to the different energy levels. For the remaining food and drink groups that were not included in the dietary guidelines, the intake from DANSDA 2011–2013 was maintained. If nutrient goals were not met, the modifiable elements were main food group or subgroup amounts, including/excluding certain foods.

#### 2.2.5. Step 5: Estimating Amount of Discretionary Energy

The maximum amount of energy for EDNP food and drinks was estimated as the difference between the energy content of the modeled recommended diets and the total energy requirement for each age and sex group.

In the official Danish Dietary Guidelines 2013 and the Nordic Nutrition Recommendations 2012, the goal for the total diet is that the content of added sugars and saturated fat should be less than 10% of energy. Therefore, the robustness of the estimated amount of discretionary energy was tested by examining the influence on the diet content of saturated fat and added sugars if the entire discretionary energy was used for single foods or drinks that contained a lot of saturated fat and/or added sugars such as chocolate or hard candy.

The amount of discretionary energy was also calculated for persons with a low or high level of physical activity. The calculations were conducted on the assumption that the proportion of the energy requirement that can be used as discretionary energy at the different activity levels was the same as for a moderate physical activity level.

## 3. Results

### 3.1. Nutrient Profiling Model

A decrease in the model scores of 2 points was seen for every 5th percentile from the 95th percentile (score = 90) down to the 30th percentile (score = 61). From the 25th percentile (score = 54) and down, the score dropped by 7 points for every 5th percentile, indicating a substantial drop in the nutrient value of the foods. In order not to overestimate the number of EDNP food and drinks, the nutrition expert group selected a score of 40 as the cut-off value. All foods and drinks with a score above 40 were classified as core foods, while food and drinks with a score ≤40 were classified as EDNP food and drinks.

Some food and drinks were manually reclassified from EDNP food and drinks to core food and drinks and vice versa. This was done to make it easier to communicate what types of food and drinks are classified as EDNP to the Danish population, but also to make the classification less sensitive to changes in the nutrient composition of single products. Some fatty meat products, cheese products, sweet breakfast cereals, and a single fish product were moved to core foods, where the most foods were placed. The reverse was also the case, i.e., some cakes, desserts, and ice cream products were classified as core foods by the nutrient profiling model and were moved to the EDNP food and drinks by the nutrition expert group.

Four EDNP food groups were reclassified as core food groups and one core drink group as EDNP drinks by the nutrition expert group independent of the model scoring: (1) vegetable oils were reclassified as a core food group due to their beneficial fatty acid composition, aligning with the official Danish Dietary Guidelines; (2 + 3) visible sugars and fat spreads were reclassified as core food groups as they are primarily used as ingredients and only consumed in small amounts, e.g., on bread or in coffee/tea; (4) water was reclassified as a core drink because it contributes to the fluid balance without providing unnecessary calories, aligning with the official Danish Dietary Guidelines [14]; (5) artificially sweetened drinks (ASD) were reclassified as EDNP drinks.

The food scoring interval of the 1482 foods was from 18 (chocolate bar) as the lowest value to 96 (Brussels sprouts) as the highest. Most foods scored between 60 and 90 (core foods), while approximately 14% scored 40 or less (EDNP foods). For drinks, a significantly larger proportion (40%) scored 40 or less (EDNP drinks).

Vegetables (including legumes), fruits and berries, composite dishes, fish, and potatoes scored the highest. Fat- and sugar-rich foods such as candy, chocolate, sugar-free candy and chocolate, cakes, biscuits, snack bars, ice cream, desserts, and snacks (chips, pork rinds, etc.) scored the lowest (Figure 2).

Drinks such as water, tea, and coffee with and without milk scored the highest (70–89), and sugar-sweetened soft drinks, cordials, energy drinks, and sports drinks as well as alcoholic drinks (beer, wine, liquor, spirits, cider) scored the lowest (14–31) of the total 161 drinks. Food and drinks with a score below 40 were classified as EDNP food and drinks and are shown in Table A1 (Appendix B). The average energy density for EDNP foods was 1670 kJ/100 g (including sugar-free candy and chocolate), while for core foods, it was 845 kJ/100 g (including fats such as vegetable oils, butter, and fat spreads, sugar, and honey). The average energy density for EDNP drinks was 360 kJ/100 g (including sugar-free and light variants), while for core drinks, it was 135 kJ/100 g.

#### Model Validation—Sensitivity and Specificity Analysis

As shown in Table 1, the final profiling model classified 1180 core foods correctly as core foods and 75 core foods incorrectly as EDNP foods. The model classified 153 EDNP foods correctly as EDNP foods and 74 EDNP foods incorrectly as core foods. In total, 149 foods were incorrectly classified compared to the expert-adjusted model. The sensitivity of the model is 94%, which means that approximately 9 out of 10 core foods are correctly classified as core foods. The specificity of the model is 67%, which means that almost 7 out of 10 EDNP foods are correctly classified as EDNP foods. The test result from the ROC procedure of 0.93 (*p* < 0.001) indicates that the model is good at classifying foods as core and EDNP foods.

As shown in Table 2, the model classified 78 core drinks correctly as core drinks and 2 core drinks incorrectly as EDNP drinks. The model classified 59 EDNP drinks correctly as EDNP drinks and 22 EDNP drinks incorrectly as core drinks compared to the expert-adjusted model. In total, 24 drinks were incorrectly classified. The sensitivity of the model is 97%, which means that almost 10 out of 10 core drinks are correctly classified as core drinks. The specificity of the model is 73%, which means that 7 out of 10 EDNP drinks are correctly classified as EDNP drinks.

The test result from the ROC procedure of 0.89 (*p* < 0.001) indicates that the model is good at classifying drinks as core and EDNP drinks.

Based on the ROC and validity tests, a cut-off value of 53–54 would have been optimal for both foods and drinks in the nutrition profiling model (specificity and sensitivity of 80–85%) and might have saved some of the reclassification carried out by the nutrition expert group.

### 3.2. Estimating the Amount of Discretionary Energy in the Recommended Diets

The recommended diets are illustrated in Table 3. The modeled recommended diets do not meet the RI for vitamin D for any age and sex group. Likewise, all age and sex groups exceed the maximum recommended sodium intake. Furthermore, iron was a challenge in the youngest age group and for females of childbearing age who have an increased need. This was handled by increasing the proportion of green vegetables (spinach, kale, and broccoli) in the “coarse vegetables” food group as well as whole grains (rye bread, oatmeal, and whole-grain pasta). In addition, there were challenges with selenium in the oldest age group. This was handled by increasing the amount of liver pâté slightly.

As shown in Table 4, discretionary energy tested for robustness constituted 4–6% of the energy requirements in all age and sex groups. Four- to six-year-olds have a low energy requirement, and therefore the lowest amount of discretionary energy available. Fourteen- to sixty-year-old males have the highest energy requirements and thus the largest amount of discretionary energy. EDNP food and drinks were communicated in two portion sizes: small portions for 4–13-year-olds and 61–75-year-olds with the lowest energy requirements, and regular portions for 14–60-year-olds with the highest energy requirements. The portions are based on the most frequently chosen portion sizes in DANSDA 2011–2013 of EDNP foods. A small portion corresponds to 450 kJ and a regular portion to 700 kJ.

#### 3.2.1. Sweet Drinks

Zero intake or as low an intake as possible of sugary drinks is recommended internationally to prevent the development of obesity and the risk of obesity-related diseases [3,4,7,23]. Therefore, it seems sensible to set a maximum limit for sweet drinks so they only constitute a small part of discretionary energy. In the present study, the maximum limit was set to ¼ liter per week for 4–6-year-olds, ⅓ liter for 7–9-year-olds and 61–75-year-olds, and ½ liter per week for 10–60-year-olds (Table 4).

#### 3.2.2. Robustness Test and Physical Activity Level

The robustness tests showed that a predominantly one-sided choice of fatty discretionary foods such as chocolate makes it difficult to comply with the nutrition recommendation for saturated fat if the amount of discretionary energy is too large. As the amount of discretionary energy should comply with nutrition recommendations, we believe it is sensible to convey the robustness-tested amount of discretionary energy to the public. In Table 4, discretionary energy is also shown as weekly portions of 450 and 700 kJ.

Physical activity does not significantly increase the amount of discretionary energy. By going from a low to a high level of physical activity, there is room for approximately one extra portion of EDNP foods or drinks (data not shown).

## 4. Discussion

The results show that the nutrient profiling model classified around 70% of 1643 foods and drinks correctly as EDNP compared with an expert-adjusted model. The specificity of foods was 67% and the sensitivity 94%, while the specificity of drinks was 73% and the sensitivity 97%. Candy, chocolate, cake, cookies and biscuits, sweet and salty snacks, sugar-sweetened and artificially sweetened drinks, and alcoholic drinks were classified as EDNP food and drinks. The estimated amount of discretionary energy for EDNP food and drinks was 4–6% of the energy requirements for 4–75-year-old Danes.

Studies assessing healthy and less healthy food and drinks with nutrient profiling models and estimating discretionary energy in a recommended diet are sparse. Anyway, we compared our main findings with the relevant studies.

### 4.1. Nutrient Profiling Model

The authors are not familiar with other studies that have classified EDNP food and drinks with a nutrient profiling model. Quinio et al. [24] found a specificity of 40–79% and a sensitivity of 55–95% for the ability of different nutrition profiling models to classify foods that are either positively or negatively associated with healthy eating habits [24]. The specificity of 67–73% and the sensitivity of 94–97% for food and drinks in the present study are at the higher end of these intervals. Another study aimed to examine the validity of five nutrient profiling models to classify foods as healthy and less healthy: Australia/New Zealand (FSANZ), France (Nutri-Score), Canada (HCST), Europe (EURO), and the Americas (PAHO), using data from the Food Label Information Program (15,342 foods and drinks) [25]. The study used a previously validated model as the reference, the Ofcom model. The agreement was assessed by the κ statistic. The results showed large variations in the agreement from κ = 0.26 to 0.89. Both FSANZ and the Nutri-Score model showed very good agreement with the Ofcom model. However, all three models included exactly the same seven nutrients, although scores were calculated differently. Other nutrient profiling models have assessed the nutrient density of individual foods using a few nutrients. Positive scores are typically estimated from protein, dietary fiber, and micronutrients such as K, Ca, Fe, and vitamin D. Negative scores are typically estimated from saturated fat, total or added sugars, and sodium [26]. Differences in the study design of the above studies make comparability with the current study difficult. In the present study, we used 20 qualifying and 4 disqualifying nutrients because our aim was to capture the nutrient density of many different foods and drinks that Danes consume.

Although a nutrient profiling model may be useful to classify food and drinks, it cannot stand alone. It is necessary to make pragmatic decisions and exceptions for some food and drinks, such as ASD.

ASD were placed among EDNP drinks, although the profiling model classified them as core drinks. The available evidence does not directly support the role of ASD in promoting or preventing weight gain or metabolic abnormalities [27]. Evidence on the impact of ASD on child health is even more scarce and inconclusive than in adults [28]. In accordance with the American Heart Association and Recommendations from Key National Health and Nutrition Organizations in the United States, it is recommended that children (and adults) limit their intake of ASD along with sugary drinks [29,30]. ASD may take the place of more nutritious drinks, which contribute to a healthy and varied diet. There is a lack of evidence to characterize the association of intake of ASD with nutrient intake and dietary patterns, and metabolic outcomes, in this population. Finally, ASD may harm dental health. ASD are therefore not considered as healthy alternatives to water [29]. Therefore, we decided to classify ASD as EDNP drinks.

Furthermore, ingredients such as fats and sugars were also reclassified as core foods after the scoring with the nutrient profiling model because they are regarded as ingredients in food preparation in Danish food culture. We believe the nutrition profiling model and the approach to estimate the amount of discretionary energy may be useful for other nutrition researchers when defining EDNP food and drinks. However, exceptions and reclassifications of some food and drinks will most likely be necessary when adopting the approach for other food cultures.

### 4.2. Estimated Amount of Discretionary Energy for EDNP Food and Drinks

The 4–6% of the energy in a recommended diet that may be used on EDNP food and drinks is consistent with the WHO’s recommendation and the EFSA’s opinion on dietary sugars, emphasizing the additional health benefits of reducing the intake of free sugar to less than 5% or as low as possible of the total energy intake [31,32]. Recently, the 2020 Dietary Guidelines Advisory Committee has suggested that less than 6% energy from added sugars is more compatible with a dietary pattern that is nutritionally adequate [3].

Denmark, the USA, and Australia have used a similar diet modeling and a comparable method to estimate the amount of energy for EDNP food and drinks [11]. However, there are significant differences in the definitions of EDNP food and drinks and in the amount of discretionary energy in the recommended diet between the three countries. In the USA, the discretionary energy is presented as an amount of saturated fat and added sugars, and not as EDNP foods and drinks. Moreover, the USA has modeled the recommended diets with the most nutrient-dense and low-energy-dense food and drinks available [3]. Therefore, the amount of discretionary energy in the recommended diet is higher in the USA than in Denmark (6–18% vs. 4–6%) [5]. In Australia, EDNP food and drinks also include sweetened condensed milk, jam, and honey, processed meats, fast food and deep-fried foods, cream, butter, and blended fat spreads. When such foods are removed from the core diet and included as EDNP foods, the recommended diet becomes more nutrient-dense and less energy-dense. The amount of discretionary energy in the recommended diet is therefore also higher in Australia than in Denmark (4–14% vs. 4–6%) [10].

The authors decided to communicate discretionary energy as portions of EDNP food and drinks where each portion corresponds to a fixed amount of energy. The Dietary Approaches to Stop Hypertension (DASH) diet is an example of a recommended diet in the USA [3,9]. In the DASH diet, discretionary energy is also communicated as a number of weekly portions. However, the DASH diet’s weekly portions are smaller than the Danish portions of EDNP food and drinks (250–400 vs. 450–700 kJ). More discretionary energy is therefore available in the Danish recommended diet than in the DASH diet [33,34].

The safest amount of discretionary energy in the recommended diet is the one tested for robustness. Unlike for fruit and vegetables, fish, and whole grains, there is no recommendation to vary the intake of candy, chocolate, soft drinks, or alcoholic drinks; therefore, it is important that the amount of discretionary energy is robust, so it is possible to choose favorable EDNP food and drinks freely.

The disease burden in Denmark shows that obese people have a shorter life expectancy, higher sickness absence, and a greater disease burden [35]. Therefore, major health benefits can be achieved by limiting the intake of EDNP food and drinks to avoid overconsumption of energy in the population.

### 4.3. Strenghts and Limitations

It is a strength that the nutrient modeling of the recommended diet was based on real intakes of food and drinks consumed by the Danish population, making the recommended diets more credible.

There are several limitations in the present study. The approach and method of the nutrient profiling model may appear complex. However, the nutrient profile scoring is based on simple calculation rules with no complex statistics that are easily performed in an Excel sheet or in statistical software programs. The scoring of nutrients and energy density in the model follows simple principles and calculation rules, as shown in Appendix A. We used a direct method to calculate the score of a food or drink based on the content of nutrients and energy relative to Nordic reference values for nutrient and energy intake.

It may also be seen as a limitation that expert judgement overrules the classification by the nutrient profiling model of some foods and drinks. However, exceptions were also made by other well-known nutrient profiling models such as the Nutri-Score, where alternative nutrient criteria and/or pre-determined cut-off scores are considered for certain cheeses, fats, and beverages [19,26]. In other nutrient profiling systems, there also exists an element of subjective judgement using pre-determined cut-off scores, nutrients, and/or the food groups included in the model [26].

As the present nutrient profiling model could be used to inform dietary guidelines, it is also important to align with other foods and drinks highlighted in the dietary guidelines such as water and plant-derived fats. A point for consideration for future development of the model could be how to handle exceptions. One suggestion could be to exclude exceptions, e.g., according to dietary guidelines before scoring. Still, this is also a subjective decision.

The values in the food composition tables used for the nutrient profiling model are average values with large variation, which may result in some inaccuracy both in the nutrient profiling and in the suggested amounts in the recommended diets. Furthermore, missing values for vitamins were found, especially for alcoholic drinks and artificially sweetened candy. However, since alcoholic drinks and artificially sweetened candy contain few vitamins, they were kept in the model. We also applied the same weighting of all 24 nutrients which may be a weakness in the model, although it is not possible to do it in a different way based on the existing knowledge. The model will not necessarily be suitable for classifying fortified foods in their present form, and these should probably be assessed separately. Another limitation was how the cut-off value was set. The cut-off was selected by the nutrition expert group, guided by the scores, the curve of the scores, and the types of food groups below and above the value. The ROC procedure showed that the cut-off of 40 was set too low as a score around 53 probably would have reduced the reclassification of food and drinks. However, there was no objective way of setting the most accurate cut-off. The setting of upper and lower cut-off values for energy density is also a limitation. As there is no common or standard definition of the energy density of high-energy-dense foods, we choose a cut-off value published by the World Cancer Research Fund/American Institute for Cancer Research in 2007 [18]. For drinks, we did not find any published cut-off values for high-energy-dense drinks. Therefore, we found inspiration in the Nuti-Score model and used this as the cut-off [19]. The validation of the model was performed against the expert-modified classification, which is also subjective.

## 5. Conclusions

The present study is the first to classify food and drinks that may be defined as EDNP according to a nutrition profiling model and nutrition expert judgement. Candy, chocolate, cake including cookies and biscuits, sweet and salty snacks, and sugar-sweetened and artificially sweetened drinks, as well as alcoholic drinks, were classified as EDNP food and drinks. Our study may be viewed as a starting point and inspiration towards an international discussion on how to define EDNP food and drinks. Furthermore, the amount of discretionary energy that may be used for EDNP food and drinks in a recommended diet was estimated to be 4–6% of the energy requirements for 4–75-year-old Danes.

It seems prudent to have special attention on EDNP food and drinks in dietary guidelines and in future public health initiatives to avoid overconsumption of energy and prevent overweight. EDNP food and drinks classified by the nutrition profiling model and the estimated amount of discretionary energy may be used to inform dietary guidelines. If people are to limit their intake of EDNP food and drinks, they must know what is classified as discretionary food and drinks, and the amount of these food and drinks that may be included in a recommended diet.

## Figures and Tables

**Figure 1 nutrients-14-01477-f001:**
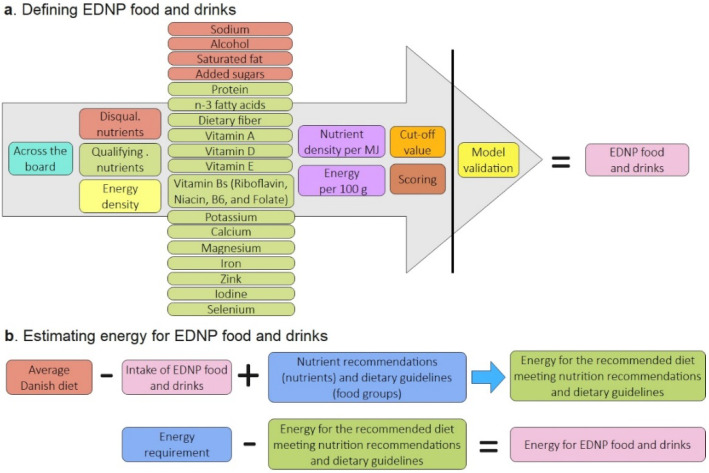
(**a**) An across-the-board nutrient profiling model including nutrient and energy density was used to define energy-dense, nutrient-poor (EDNP) food and drinks. The visual model was adapted from Verhagen and Van den Berg [13]. (**b**) The average Danish diet without EDNP food and drinks was used to model recommended diets that meet the dietary guidelines and nutrient recommendations. The amount of energy for EDNP food and drinks was defined as the energy requirement for different age and sex groups minus the energy needed to meet a recommended diet.

**Figure 2 nutrients-14-01477-f002:**
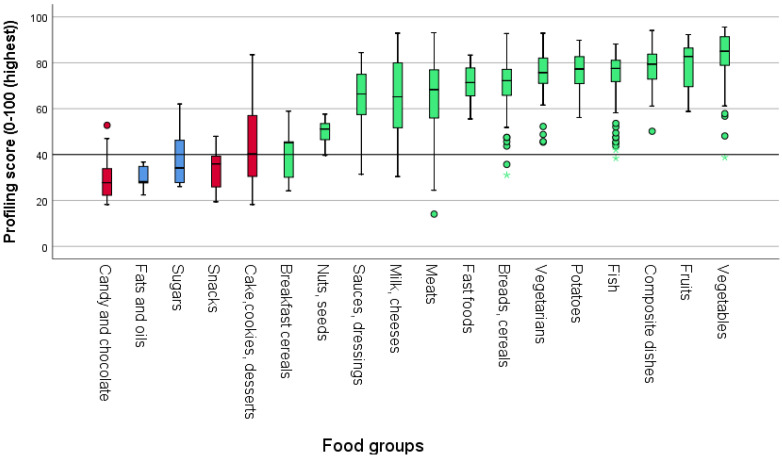
Classification of foods is divided into main food groups by the nutrient profiling model. Green = core foods; red = energy-dense, nutrient-poor foods; blue = energy-dense, nutrient-poor ingredients. ° Mild outliers based on the interquartile range (IQR): Q1 ± (1.5 × IQR). ☆ Extreme outliers based on the interquartile range (IQR): Q1 ± (3 × IQR).

**Table 1 nutrients-14-01477-t001:** Overview of correctly and incorrectly classified core and energy-dense, nutrient-poor (EDNP) foods when comparing the predicted model with the adjusted expert model. Values for sensitivity and specificity are given.

		Final Classification		
		Core Foods	EDNP Foods	Total
Predicted classification	Core foods	1180	74	1254
	EDNP foods	75	153	228
	Total	1255	227	1482
Sensitivity (%)			94	
Specificity (%)			67	

**Table 2 nutrients-14-01477-t002:** Overview of correctly and incorrectly classified core and energy-dense, nutrient-poor (EDNP) drinks when comparing the predicted model with the adjusted expert model. Values for sensitivity and specificity are given.

		Final Classification		
		Core Drinks	EDNP Drinks	Total
Predicted classification	Core drinks	78	22	100
	EDNP drinks	2	59	61
	Total	80	81	161
Sensitivity (%)			97	
Specificity (%)			73	

**Table 3 nutrients-14-01477-t003:** Modeled recommended diets for different age and sex groups meeting the official Danish Dietary Guidelines 2013.

	4–6 Year (*n* = 203)	7–9 Year (*n* = 218)	10–13 Year (*n* = 269)	Males 14–60 Year (*n* = 1206)	Females 14–60 Year (*n* = 1289)	61–75 Year (*n* = 761)
Unprocessed red meat, g/week	185	225	350	350	350	350
Processed red meat, g/week	0	0	0	0	0	0
Fish, g/week	210	250	350	350	350	350
Fatty fish, g/week	120	140	200	200	200	200
Lean fish, g/week	90	110	150	150	150	150
Vegetables, g/day	200	250	300	300	300	300
Vegetables coarse, g/day	100	125	150	150	150	150
Vegetables fine, g/day	100	125	150	150	150	150
Fruit, g/day	200	250	300	300	300	300
Fruit juice max., g/day	65	80	100	100	100	100
Whole grain, g/day	45	53	68	88	71	67
Milk and dairy products, g/day	250	250	250	250	250	250
Cheese, g/day	10	15	17	30	25	20
Nuts, g/day	20	20	30	30	30	30

**Table 4 nutrients-14-01477-t004:** Overview of energy requirements and the energy content of the recommended diets, and discretionary energy for 4–75-year-old Danes.

Age and Sex Groups	4–6 Year (*n* = 208)	7–9 Year (*n* = 218)	10–13 Year (*n* = 269)	Males 14–60 Year (*n* = 1206)	Females 14–60 Year (*n* = 1289)	61–75 Year (*n*= 761)
PAL (moderate physical activity level)	1.57	1.57	1.73	1.73/1.60 *	1.73/1.60 *	1.60
Energy requirement (MJ/day)	6.0	7.1	9.1	11.7	9.4	8.9
Energy content in the recommended diets (MJ/day)	5.8	6.8	8.5	11.0	8.9	8.5
Discretionary energy after robustness test (MJ/week)	1.6	2.3	3.8	5.0	3.5	2.6
Percent discretionary energy of the energy requirements (%)	4	5	6	6	5	4
Portion size (kJ)	450	450	450	700	700	450
Max. number of weekly portions	4	5	8 **	7	5	6
Max. amount of weekly sweet drinks *** (cl/week)	25	33	50	50	50	33
Max. number of weekly alcoholic standard drinks ****	0	0	0	11 (14–17 year = 0)	7 (14–17 year = 0)	6

* 1.73 for 14–17-year-olds and 1.60 for 18–60-year-olds. ** Maximum weekly portions were reduced from nine to eight for 10–13-year-olds to make the transition between age groups for children and young adults smaller in the number of weekly portions. *** Sugar-sweetened and artificially sweetened drinks such as sodas, juices, iced tea, energy drinks, and sports drinks (1 small portion = 25 cl; 1 regular portion = 50 cl). **** Alcoholic drinks may be included for 18+ if kept within the maximum limits of standard drinks (14 a week for males and 7 a week for females). One standard drink is approximately 450 kJ (beer ordinary (33 cl), gold beer 5.6% (25 cl), strong beer 7% (22 cl), wine (12 cl), liqueur wine (8 cl), spirits (4 cl)). **** If 18–60-year-old males choose to drink 11 standard drinks a week, all discretionary energy will be spent on alcohol. If 18–60-year-old females choose to drink seven standard drinks a week, there is a half portion left. If 61–75-year-olds choose to drink six standard drinks a week, all discretionary energy is spent on alcohol.

## Data Availability

In accordance with Danish law and GDPR, the data used in this study can only be accessed through the servers at the Technical University of Denmark and may also require a Disclosure Declaration. Access and a Disclosure Declaration can be requested if the applicant fulfills the criteria for access. The Technical University of Denmark can be contacted by e-mail: apbj@food.dtu.dk.

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
