# Peer review of "Defining Energy-Dense, Nutrient-Poor Food and Drinks and Estimating the Amount of Discretionary Energy"

_nutrients, 2022, doi:10.3390/nu14071477_

Round 1
Reviewer 1 Report
This study aimed to develop a nutrient profiling model for classifying EDNP food and drinks and to estimate the amount of discretionary energy for those foods. The manuscript is hard to follow in the current form largely because many author-defined terms are used without proper introduction. Also, the manuscript is very long- I believe the whole process can be streamlined. The application of nutrient profiling model seems like a good idea, but there are a lot of exceptions and adjustments along the process, which makes the application of the same methodology to other contexts very hard. Probably that’s why there’s no internal definitions for ENDPs yet. I think the authors may need to narrow down the purpose of this paper- either focusing on setting up the standard procedure or listing EDNP foods and drinks for contemporary Dutch people. Some minor comments are as follows:
Line 113: On what basis were these nutrients and food components selected? Please note that fiber and added sugars are not often considered as ‘nutrients,’ which is contradictory to the statement in line 98.
Line 116: If the DV for “the most demanding person” is used, are these only applicable to adults given that children have very different requirements?
Line 162: Please introduce the concept of ‘core foods’ early in the paper.
Line 181: Please describe the rationale for applying the separate method to classify this specific set of foods. Were these foods not categorized as core or EDNP foods when scored with the nutrient profiling model?
Line 192: How’s the final nutrient profiling model different with the expert adjusted model? Please introduce the expert adjusted model or the expert model later on somewhere.
Reviewer 2 Report
Manuscript title:
Nutrients-1642781 - Defining energy-dense, nutrient-poor food and drinks and 2 amount of discretionary energy
|
Títle |
|
|
Is it understandable and concise? |
( x ) Yes ( ) Not |
|
Reflects the content? |
( x ) Yes ( ) Not |
|
Abstract |
|
|
It includes: objectives, methodology, key findings and conclusions? |
( ) Yes ( x ) Not |
|
Introduccion |
|
|
The investigation was carried out in a suitable theoretical structure? |
( ) Yes ( x ) Not |
|
Clear leaves the questions you want to answer and objectives of the work? |
( ) Yes ( x ) Not |
|
The cited references are current and relevant? |
( ) Yes ( x ) Not |
|
Methods |
|
|
The methods presented are appropriate to achieve the proposed objectives? |
( ) Yes ( x ) Not |
|
The selection and composition of the sample are adequately described? |
( ) Yes ( x ) Not |
|
The data collection process and the tools used are described clearly? |
( ) Yes ( x ) Not |
|
The statistical analysis and the research design appropriate? |
( ) Yes ( x ) Not |
|
Results |
|
|
The presentation of the results clear? |
( ) Yes ( x ) Not |
|
The main results are highlighted without the inclusion of interpretation and comparisons? |
( ) Yes ( x ) Not |
|
The results evaluate the proposed objectives? |
( ) Yes ( x ) Not |
|
Tables and figures are properly numbered, labeled and explained? |
( ) Yes ( x ) Not |
|
Discussion and Conclusion |
|
|
The results are discussed based on the literature? |
( ) Yes ( x ) Not |
|
Author's interpretations show the safety and soundness? |
( x ) Yes ( ) Not |
|
The limitations of the work are presented? |
( x ) Yes ( ) Not |
|
The conclusions of the study are presented? |
( ) Yes ( x ) Not |
|
The conclusions respond to the objectives? |
( ) Yes ( x ) Not |
General comments:
Title
Are presented satisfactorily.
Abstract
The abstract does not conform to the journal's template. Please review.
The Abstract does not bring numerical evaluations and calculations that were made, that is, that of the manuscript as a whole.
The study problem is not very clear. It would be good in the conveniences of the finds.
Keywords are not in accordance with those described in health science. Please adjust.
Introduction
The introduction is too long, and it doesn't present the transition from general to specific very well. The introduction should have between three and five paragraphs, starting with the epidemiological context, moving on to the relationship between this and what is intended to be studied, then indicating the problem/gap of study, objectives and finalizing the hypotheses raised.
The academic support pillars of the text end up not being very clear, what would be studied in relation to the epidemiological problem presented at the beginning of the introduction.
The problem is not properly limited and presented. It is suggested that some hypotheses be presented to be answered by the study.
Methods
It should present more clearly the design of the study. The CONSORT or timeline should be presented in order to get a better view of the study design.
There are some points in the methodology that would be more linked to the introduction than to the methodology.
It is suggested that the methodology be readjusted in the following points, design, parameters to be used, that is, instruments and procedures, and later statistics.
The steps presented and the calculations performed offer an important parameter, however, it must be redone as they were placed, which ends up promoting confusion. Please adjust this.
The statistics are a little confusing, not allowing a clear view of what was done in this item of the methodology.
Results
Some findings do not find support in the methodology and statistics to justify how the result was conceived.
Some tables are actually tables having no numerical value. It would be nice to have a better description of the findings in statistical terms, which is not the case.
In the alcohol item, some comments that should only be in the discussion would not be appropriate. In the result item, there should only be the results and an explanation of them, and leave other considerations for the discussion.
Discussion
It should reaffirm the objectives and start discussing the results in the chronological order that appear in the item results.
However, the discussion was limited to focusing on the justification of the findings and not providing a more exhaustive discussion, which would lead to a more critical view of the findings.
Conclusion
Are presented satisfactorily. However, practical applications should be better explored.
References
Please review the formatting of the references, and of the 38 references we have 23 current and 15 with more than five years of publication.
Overview
The manuscript presented addresses a relevant research topic.
It would be advisable to do a general review.
Some points, as the authors' contribution is not in accordance with the journal's norms.
Round 2
Reviewer 2 Report
Bearing in mind that the adjustments were all made, within what was pointed out, I consider that the manuscript is in a condition to be published.
Author Response
The authors are pleased to be informed by reviewer 2 that the the revised manuscript is in a condition to be published.